# Variant surface glycoprotein density defines an immune evasion threshold for African trypanosomes undergoing antigenic variation

Jason Pinger[1,2], Shanin Chowdhury[1] & F. Nina Papavasiliou[1,3]

*Trypanosoma brucei* is a protozoan parasite that evades its host's adaptive immune response by repeatedly replacing its dense variant surface glycoprotein (VSG) coat from its large genomic *VSG* repertoire. While the mechanisms regulating *VSG* gene expression and diversification have been examined extensively, the dynamics of VSG coat replacement at the protein level, and the impact of this process on successful immune evasion, remain unclear. Here we evaluate the rate of VSG replacement at the trypanosome surface following a genetic *VSG* switch, and show that full coat replacement requires several days to complete. Using in vivo infection assays, we demonstrate that parasites undergoing coat replacement are only vulnerable to clearance via early IgM antibodies for a limited time. Finally, we show that IgM loses its ability to mediate trypanosome clearance at unexpectedly early stages of coat replacement based on a critical density threshold of its cognate VSGs on the parasite surface.

---

[1] The Rockefeller University, Laboratory of Lymphocyte Biology, 1230 York Avenue, New York, NY 10065, USA. [2] The David Rockefeller Graduate School, 1230 York Avenue, New York, NY 10065, USA. [3] Division of Immune Diversity, German Cancer Research Center, Im Neuenheimer Feld 280, Heidelberg 69120, Germany. Correspondence and requests for materials should be addressed to J.P. (email: jpinger@gmail.com) or to F.N.P. (email: n.papavasiliou@dkfz-heidelberg.de)

The protozoan parasite *Trypanosoma brucei*, a causative agent of human and animal trypanosomiasis, lives extracellularly within its host and evades host immunity through antigenic variation of its variant surface glycoprotein (VSG) coat. Each parasite expresses one *VSG* gene at a time from a genomic repertoire of ~2000[1], and is densely coated with ~$10^7$ VSGs[2]. During infection, the host develops potent VSG-specific antibodies (Abs) that mediate trypanosome clearance, but a minority of parasites evade clearance by switching expression to antigenically distinct *VSGs*[2, 3]. These switched parasite populations then expand within the host until they are cleared, after which additional populations expressing distinct VSGs emerge again. This cyclical process results in characteristic waves of parasitemia occurring at ~5–8 day intervals during infection, with parasite suppression in sync with and mediated by the development of repeated primary, VSG-specific Ab responses[3–7].

Most analyses of antigenic variation in *T. brucei* have focused on genetic factors regulating *VSG* expression and diversification, but protein dynamics also influence the host–pathogen interface and successful immune evasion. Following a genetic *VSG* switch, trypanosomes must replace their entire VSG coat. During this period, trypanosomes simultaneously display both pre- and post-switch VSGs on their surface, a phenomenon that has been observed in infection isolates[8]. This coat replacement process is critical for the survival of recently switched cells because initial VSGs remain targets for the escalating host Ab response, but the dynamics of VSG replacement remain poorly understood. VSG half-life measurements suggest that initial VSGs may persist on the surface of genetically switched trypanosomes for several days[9, 10]. However, this estimate assumes that VSG turnover is identical in recently switched and non-switched trypanosomes, an assumption which has not been experimentally validated due to the low switching frequencies observed in lab adapted

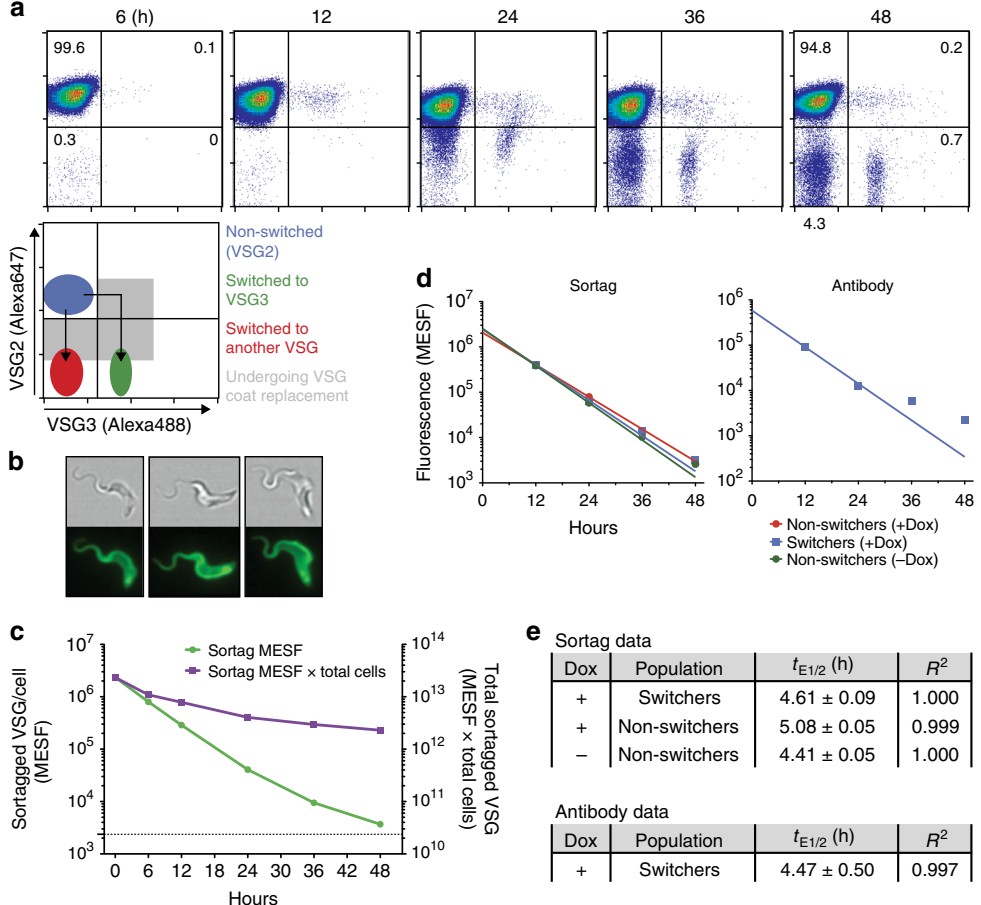

**Fig. 1** Determining the rate of VSG loss in switching and non-switching trypanosomes. **a** Δ70 trypanosomes undergoing VSG coat replacement following a genetic *VSG* switch. Cells were induced to switch at time $t = 0$ h. VSG2 is the initial, pre-switch VSG, while VSG3 is one of many possible post-switch VSGs. Percentage of cells in each quadrant are shown at 6 and 48 h. **b** Live KI-VSG2[STa] trypanosomes 12 h after sortagging with the fluorophore 5-FAM. **c** Loss of sortagged VSGs over time in KI-VSG2[STa] trypanosomes. Population mean fluorescence intensity (MFI) values, measured via flow cytometry, were converted to absolute units (Molecules of Equivalent Soluble Fluorochrome, MESF)[44]. Sortag MESF (plotted on left *Y* axis) is the mean amount of sortagged VSG remaining per cell. Dotted line is associated with the left *Y* axis and represents background MESF of non-sortagged cells. Sortag MESF × total cells (plotted on right *Y* axis) is the total cell-associated, sortagged VSG remaining in the population. This value was calculated by subtracting background MESF from the sortag MESF value and multiplying by the total number of cells in the population. **d**, **e** Calculation of rates of VSG loss ($t_{E1/2}$). Δ70[STa] cells were fluorescently sortagged, then either induced to switch (+Dox) or not induced (−Dox) at $t = 0$ h. Samples were taken for flow cytometry analysis over a 48 h period. "Switchers" were identified as those cells with lowered VSG2 levels (via Ab staining), while "non-switchers" retained full VSG2 levels (Supplementary Fig. 1c). For **d**, raw MFI values for Ab staining and sortag levels were converted to MESF units (as in **c**, see Methods). Lines on plots show non-linear regression analyses. **e** $t_{E1/2}$ calculated for indicated populations (value ± SE) and goodness of non-linear regression fit ($R^2$). Data in **a–c** are representative of at least three replicates of each experiment. $t_{E1/2}$ values (**d**, **e**) were calculated from one such experiment. One KI-VSG2[STa] and one Δ70[STa] clone were used for these experiments

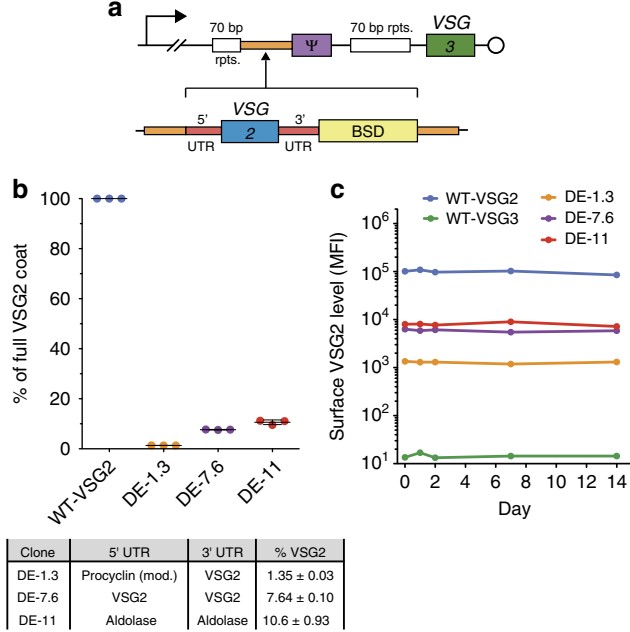

**Fig. 2** Dual-expressor clones stably express two VSGs at different ratios. **a** Construction of dual-expressor (DE) clones. (Top) Diagram of the active VSG expression site (Bloodstream Expression Site 1, BES1) of the parental cell line shows the promoter (bent arrow), regions of repetitive DNA (70 bp rpts.), a VSG pseudogene (ψ), the telomere (circle), the active VSG (VSG3), and an intergenic region unique to BES1 (orange). (Bottom) The VSG2 insertion constructs contain a Blasticidin resistance cassette (BSD) and a VSG2 gene flanked by 5′ and 3′ UTRs that are unique to each construct. These insertions are targeted to BES1 by homology to the indicated intergenic region. **b** Analysis of the level of surface VSG2 present on DE clones. Raw fluorescence intensity values (population mean fluorescence intensity (MFI) via VSG2 Ab staining at saturating concentration, see Supplementary Fig. 2a) were converted to absolute units (MESF). VSG2 level was calculated as a percentage of the MESF value of a full VSG2 coat (WT-VSG2). Table indicates the 5′ and 3′ UTRs used and the resulting percentage of surface VSG2 expressed for each clone. Individual data points are technical replicates using one clone of each type, compiled from independent experimental replications on three occasions. Bars and % VSG2 values represent mean ± SD. **c** Stability of surface VSG2 levels expressed by DE clones (population MFI, VSG2 Ab staining) in culture

## Results

**Trypanosomes do not expedite VSG turnover after a VSG switch.** To examine the possibility that trypanosomes expedite VSG turnover immediately post-switch as an immune evasion strategy, we first compared VSG turnover in recently switched and non-switched trypanosomes (Fig. 1). We employed a quantitative flow cytometry approach using a recently described transgenic cell line[13] with heightened switching capability in conjunction with a novel VSG-labeling strategy. The transgenic line (Δ70) has an I-SceI restriction site immediately upstream of the active VSG, and a tet-inducible I-SceI gene at another genomic location. In the absence of doxycycline (Dox), Δ70 cells do not switch at high frequency (Supplementary Fig. 1a). Dox induction initiates a double-strand DNA break near the initially active VSG (VSG2, a.k.a. Lister 427-2, VSG221), the repair of which yields a high percentage of trypanosomes ($\sim10^{-1}$–$10^{-2}$, varying by clone and experiment) switching exactly once to another VSG. Using this cell line and Abs recognizing pre- and post-switch VSGs, we observed VSG coat replacement in recently switched cells via flow cytometry (Fig. 1a). To examine VSG turnover in non-switched cells that maintain consistent VSG2 levels, we developed a direct VSG labeling strategy utilizing the enzyme Sortase A. Sortase A covalently links molecules containing a peptide "sorting motif" (LPXT[GG/AA]) to molecules containing an N-terminal $(Gly)_3$ or $(Ala)_2$ sequence in a process known as "sortagging"[14]. We engineered an altered VSG2 with an extended N terminus ($VSG2^{STa}$) that can serve as a substrate for the sortagging reaction. Incubating live trypanosomes expressing $VSG2^{STa}$ (KI-$VSG2^{STa}$) with purified sortase and fluorophore-sorting motif conjugates effectively ligated these fluorophores to surface VSGs (Fig. 1b, illustrated in Supplementary Fig. 1b). Clones expressing $VSG2^{STa}$ showed no difference in growth rate compared to clones expressing unaltered VSGs, and fluorescent labeling of $VSG2^{STa}$ on the surface of live cells also did not affect cell growth (Supplementary Fig. 1c). Decrease in the fluorescence intensity of sortagged trypanosome populations cultured over time indicated turnover of these labeled VSGs (Fig. 1c).

We then inserted the sortaggable $VSG2^{STa}$ into the VSG expression site of the high switching Δ70 cell line, replacing the active VSG2 gene (Δ$70^{STa}$). This allowed us to estimate the rate of VSG loss in both switched and non-switched cells by quantifying decreases in fluorescence intensity from Ab staining and sortagging over time (Fig. 1d, e; Supplementary Fig. 1d). Because flow cytometry measures individual cells, our VSG loss rates are calculated at the single-cell level, and therefore include contributions both from the dilution of VSGs during cell division (population doubling time, $t_D$) and from VSG shedding and degradation (VSG half-life, $t_{1/2}$). Evidence of the latter factor (loss of VSGs from the entire trypanosome population by shedding and degradation), is shown by the negative slope of the "Sortag MESF × total cells" curve in Fig. 1c. The combined VSG loss rate at the single-cell level is the rate most relevant to immune evasion, and we have termed this value the "effective VSG half-life" ($t_{E1/2}$, Supplementary Table 1). Calculated $t_{E1/2}$ values (~4.6 h on average, Fig. 1e) agreed between measurement methods, and were similar to previously published population-wide data (Supplementary Table 1)[9, 10]. Notably, $t_{E1/2}$ values were comparable for switched and non-switched cell populations, indicating that VSG turnover rates are identical in recently switched and non-switched cells. These results were not confounded by large differences in growth rate between cell populations (Supplementary Fig. 1e). Based on this $t_{E1/2}$ of 4.6 h, trypanosomes with fully replaced VSG coats would not arise until ~107 h (~4.5 days) after a genetic VSG switch (Supplementary Table 1), suggesting that switched trypanosomes and their progeny may be susceptible to immune clearance for extended periods.

trypanosome cell lines in vitro ($10^{-5}$–$10^{-6}$ cells per population doubling time[11, 12]). Furthermore, the vulnerability of trypanosomes with partially replaced coats to Ab-mediated clearance has not been directly investigated. Thus, the specific factors that allow switched trypanosomes to successfully evade the mounting Ab response are yet undetermined.

In this study, we evaluate the rate of VSG coat replacement through quantitative flow cytometry analysis. We demonstrate that trypanosomes do not expedite VSG turnover following a genetic VSG switch, and that switched parasites require several days to fully replace their coats. We then describe the generation of trypanosome clones expressing two VSGs at varied ratios, representing parasites at multiple stages of VSG coat replacement. Using these clones in in vivo infection assays, we show that trypanosomes are only vulnerable to immune clearance via early IgM Abs for an unexpectedly small fraction of the total coat replacement process. Following further IgM binding analyses and molecular modeling, we conclude that the immune evasion threshold we observe is determined by the inability of IgM Abs to bind cognate VSGs displayed at low densities on the parasite surface.

| Clone | 5′ UTR | 3′ UTR | % VSG2 |
|-------|--------|--------|--------|
| DE-1.3 | Procyclin (mod.) | VSG2 | 1.35 ± 0.03 |
| DE-7.6 | VSG2 | VSG2 | 7.64 ± 0.10 |
| DE-11 | Aldolase | Aldolase | 10.6 ± 0.93 |

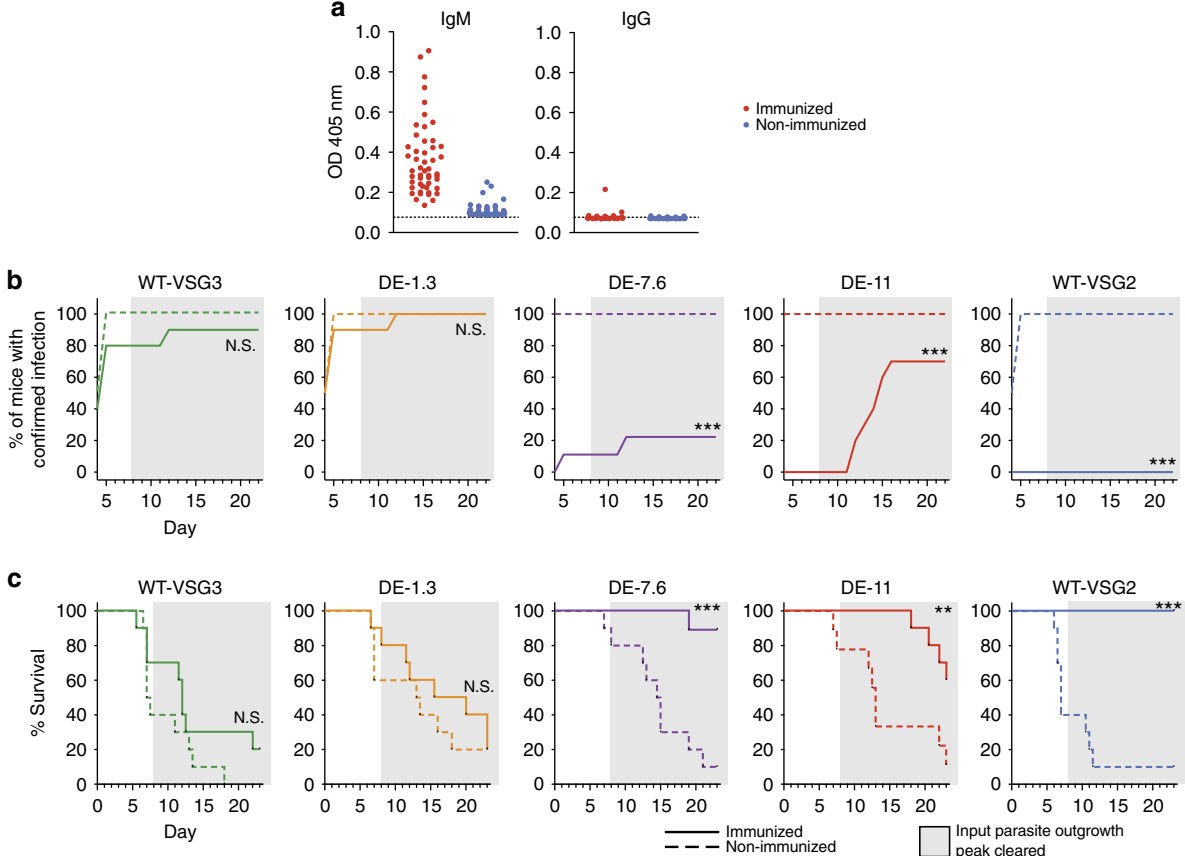

**Fig. 3** Immunization-challenge experiments with DE clones reveal a threshold of immune evasion. **a** VSG2-specific IgM and IgG Ab titers in the immunized and non-immunized mice ($n = 49$ mice/group) used for experiments shown in **b**, **c** at day 8 post immunization (combined results of three independent experiments). Data show relative titers as measured by ELISA. Dotted lines indicate average background OD in wells incubated with no serum. **b**, **c** Non-immunized mice and mice immunized against VSG2 were challenged with the trypanosome clones indicated. **b** Percentage of mice with confirmed infection over time, and **c** survival curves for immunization-challenge experiments. X axes indicate day post challenge. Gray shading indicates the period following clearance of the parasitemia peak resulting from outgrowth of input parasites. For **b**, mice were permanently scored as infected when trypanosomes were first identified via blood smear. For **b**, **c**, statistical analysis was performed using the log-rank (Mantel–Cox) test ($n = 9$ for DE-7.6 immunized and DE-11 non-immunized, 10 for other groups); ***$P < 0.001$, **$0.001 < P < 0.01$, N.S., not significant ($P > 0.05$). Data include combined results of three independent experiments (with $n = 4$ or 5 mice per experimental group per experiment). One trypanosome clone of each type was used in these experiments

**Generating trypanosomes expressing two VSGs at varied ratios**. To understand the impact of VSG coat replacement dynamics on successful immune evasion during an infection, we sought to examine the vulnerability of trypanosomes with partially replaced coats to Ab-mediated clearance. We first generated a set of dual-expressor (DE) trypanosome clones that simultaneously express two surface VSGs at various ratios, representing switched cells at intermediate stages of coat replacement (Fig. 2). In the following experiments, VSG2 and VSG3 (a. k.a. Lister 427-3, VSG224) represent pre-and post-switch VSGs, respectively. To create the DE clones, we modified a parental cell line originally expressing VSG3 (WT-VSG3) by inserting *VSG2* into the *VSG3* expression site upstream of *VSG3* (Fig. 2a), following a strategy similar to that demonstrated in previous publications[15]. Modulation of VSG2 expression was achieved by changing the 5′ and 3′ UTR sequences flanking the inserted *VSG2* gene[16, 17] (Fig. 2b), and clones generated in this manner stably expressed varied VSG2 surface levels (Fig. 2c). To ensure that our calculations of surface VSG2 expression ratios (Fig. 2b) were accurate, we performed the flow cytometry-based measurements underlying these data using a saturating concentration of anti-VSG2 Ab (Supplementary Fig. 2a). Similar measurements of VSG3 expression levels in the DE clones indicated that expression

of the introduced *VSG2* gene did not affect the amount of VSG3 present on the surface of these cells (Supplementary Fig. 2b, c). Previous studies have shown that co-expression of two VSGs does not perturb trypanosome growth or normal cellular function[15, 18], and we also observed no growth defect related to the dual VSG expression (Supplementary Fig. 2d).

**Only partial coat replacement is required for immune evasion**. During infection, T-cell independent VSG-specific IgM responses comprise the most rapid host defense against parasite proliferation[19], and non-switched trypanosomes have been shown to be cleared predominately via early arising IgM Abs[6]. We therefore developed an immunization strategy to elicit IgM in mice. Briefly, mice were inoculated with UV-irradiated VSG2-expressing trypanosomes that lacked the endogenous lipase GPI-PLC, and therefore retained intact VSG coats following irradiation[20]. This strategy elicited a VSG2-specific IgM response approximating that elicited by live trypanosomes, while significant production of class-switched (IgG) Abs was not observed during the 14 days following immunization (Supplementary Fig. 3a).

We used this immunization protocol to assess whether our DE clones could evade a VSG2-specific IgM response in vivo (Fig. 3).

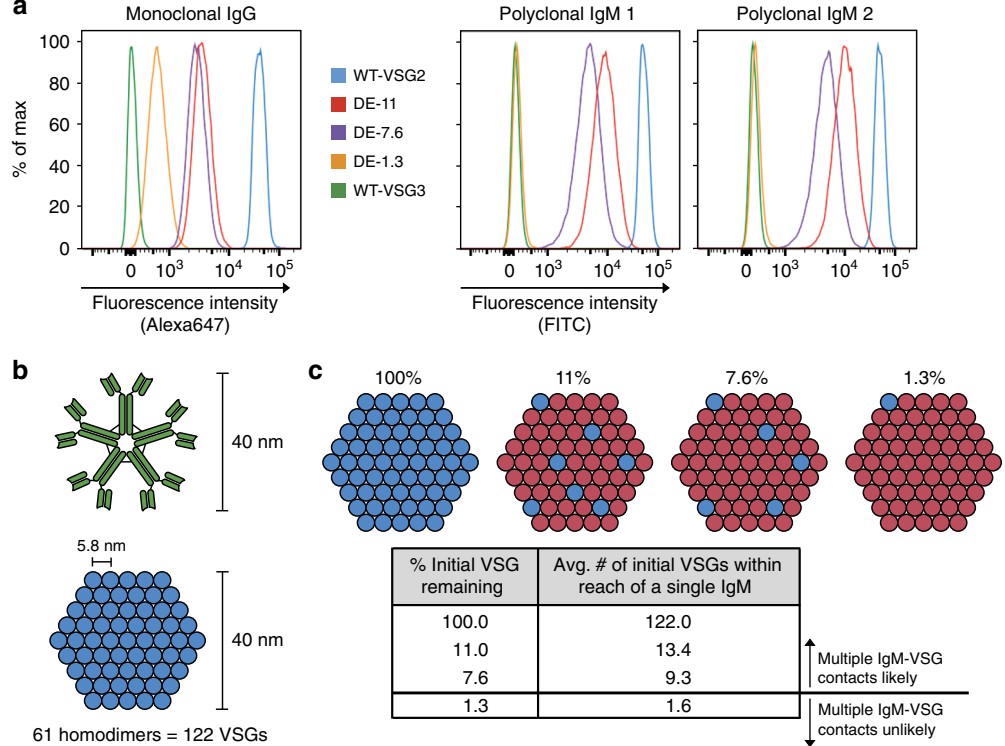

**Fig. 4** The effect of VSG density on IgM binding ability. **a** VSG2-specific Ab binding to wild-type (WT) and dual expressor (DE) clones. Monoclonal anti-VSG2 IgG Ab was used as a control to demonstrate the presence of surface VSG2 in all DE clones (left). Binding ability of two polyclonal IgM antisera is shown on the right. Data are representative of three replications of this experiment. **b**, **c** A model detailing a proposed mechanism for the observed IgM binding threshold. **b** The binding footprint of IgM. Modeling the VSG coat (bottom) as a hexagonal array of VSG homodimers with a spacing of ~5.8 nm, an IgM molecule (top) with a diameter of 40 nm could contact a maximum of 61 homodimers (122 VSGs) at a single time. **c** The average number of initial VSGs within a given IgM binding footprint at varying stages of coat replacement

Immunized and non-immunized control mice received challenge injections at day 8 post immunization, at which point the immunized mice had high IgM titers, and low or undetectable IgG titers (Fig. 3a). Challenge injections contained 100 live WT-VSG2 (expressing VSG2 only), WT-VSG3 (expressing VSG3 only), DE-1.3, DE-7.6, or DE-11 trypanosomes. Starting at day 4 post challenge, mice were monitored for survival, and blood smears were examined daily for the presence of trypanosomes. Our results clearly demonstrated a threshold surface VSG2 level required for IgM-mediated trypanosome clearance. Clones with surface VSG2 levels ≥7.6% (DE-7.6, DE-11, and WT-VSG2) were either cleared entirely or exhibited delayed infection in immunized mice (Fig. 3b), while WT-VSG3 and DE-1.3 clones were not affected by pre-immunization in terms of infection dynamics (Fig. 3b) or parasitemia levels during the first peak of infection (Supplementary Fig. 3b). VSG2 levels ≥7.6% also correlated with a survival advantage in immunized mice (Fig. 3c). The lack of evident pressure from VSG2-specific Abs on DE-1.3 trypanosomes was striking, as VSG2 proteins are abundant on the surface of these cells (~$1.3 \times 10^5$ VSG2 proteins/cell). These data suggest that switched trypanosomes escape the early IgM response once they have replaced only ~99% of their initial coat.

A more detailed analysis of these results requires separation of the infections into two stages. Data from the first ~7 days post challenge (Fig. 3b, c, unshaded regions) permits the most straightforward interpretation. The parasitemia peak representing immediate outgrowth of the input trypanosomes was uniformly detectable by day 4 or 5 post challenge in non-immunized mice. Suppression of parasitemia in immunized mice at this time must be interpreted as a direct effect of pre-immunization on the survival of the input trypanosomes (Fig. 3b), which was the

interaction we intended to examine. After this initial period, however, host–parasite interactions become more complex. If the input trypanosome peak occurred, mice either succumbed to infection immediately, or cleared the peak on days 6–8 by generating Abs in response to the challenge injection. Mortalities occurring after day 8 thus resulted from expansion of switched parasite populations expressing VSGs not present in the input injections. Notably, trypanosomes isolated from several immunized mice that showed delayed infection with DE-7.6 and DE-11 clones had indeed lost expression of both VSG2 and VSG3 (Supplementary Fig. 3c), suggesting that pre-immunization suppressed outgrowth of the input trypanosomes, but switched populations emerged following some level of early infection which had been undetectable by blood smear. Bearing these later complications in mind, the significant suppressive effects of pre-immunization on clones with surface VSG2 levels ≥7.6% are still most readily explained by the interactions of early IgM with the input trypanosomes during the first week of infection. Late interactions between maturing immune responses elicited against the input VSGs and emerging switched parasite populations may have some effect on the overall infection dynamics, but are unlikely to be the predominant factors influencing our results.

Combining these results with the VSG turnover rate measurements presented earlier, our findings permit a more complete analysis of the effect of coat replacement dynamics on successful immune evasion. Full coat replacement takes ~4.5 days, but switched cells reach 7.6% and 1.3% initial VSG (levels on either side of the immune evasion threshold) after only 17.1 and 28.8 h, respectively (Supplementary Table 1). Our data therefore suggest that although switched cells present initial VSG on their surface

for extended periods, they are only vulnerable to immune clearance for a small fraction of that time.

**IgM binding ability is dependent on VSG density**. To examine the mechanism(s) underlying this immune evasion threshold, we assessed the ability of IgM to bind to our DE clones (Fig. 4). VSG2-specific polyclonal IgM Abs were raised in two mice. Both IgM samples strongly stained DE-7.6, DE-11, and WT-VSG2 cells, but showed extremely weak or no binding to DE-1.3 and WT-VSG3 cells (Fig. 4a). To interpret this binding behavior, we developed a model predicting the number of contacts an individual IgM could make with surface VSGs present at varying densities. IgM is a pentameric, decavalent molecule whose binding capability is generally based on low-affinity, high-avidity interactions[21]. IgM is ~30–40 nm in diameter[22, 23], while the VSG coat has been modeled as a hexagonal array of VSG homodimers with a spacing of ~5.8 nm[24] (Fig. 4b). Using these parameters, we predicted the number of initial VSGs within reach of a 40 nm wide (Fig. 4c) or 30 nm wide (Supplementary Fig. 4) IgM at the coat replacement stages represented by our DE clones. We estimate that a single IgM molecule can reach multiple initial VSGs at any given moment when initial VSG composes ≥7.6% of the total surface VSG, but this is no longer true when that value reaches 1.3%. Thus, our model suggests that the observed in vivo immune evasion and in vitro IgM binding thresholds are driven by IgM's fundamental requirement for multiple antigen contacts to allow successful binding and performance of effector functions. We conclude that antigenic density, not abundance, is a key determinant of the efficacy of early IgM responses.

## Discussion

We have demonstrated here that trypanosomes require ~4.5 days to fully replace their VSG coats following a genetic *VSG* switch, and that these cells are only vulnerable to clearance by early IgM Abs for, at most, the initial ~29 h of that period. To fully evaluate the impact of these results in the context of an infection, we must also consider the timing of the genetic switch in relation to the establishment of the immune response. It remains unclear whether regulation of *VSG* switching is purely stochastic or whether environmental stimuli affect switching frequency, but the fact that switching occurs in vitro suggests that there is at least some host-independent, stochastic element to the process[25, 26]. Analyses of infection dynamics indicate that early VSG-specific IgMs only reach titers capable of clearing non-switched trypanosomes 5–6 days after the initial stimulus[6]. Trypanosomes switching very early in infection or near the beginning of waves of parasitemia may therefore fully replace their VSG coats before the IgM response becomes effective, so it is possible that infections could persist even if trypanosomes remained vulnerable for longer portions of the coat replacement process. However, the limited vulnerability window we have described certainly does impact the character of each successive wave of parasitemia. The total number of trypanosomes present in an infected host exponentially increases as a wave of parasitemia reaches its peak. Whether or not *VSG* switching is purely stochastic, the absolute number of independent switch events must also accumulate accordingly nearer to the peak. If all of the parasites that switch up to 29 h before a parasitemia peak are able to survive (as opposed to only those that switch several days before the peak), this would imply a significantly greater number of switched cells forming the subsequent wave of parasitemia, and consequently a greater diversity of VSGs expressed among those cells. A recent study conducted by our group demonstrated that VSG expression is highly diverse within individual waves of parasitemia, and that some major variants comprise large portions of the population, while some minor variants appear only transiently[3]. It is possible that high VSG diversity is necessary for selection of the few variants that will be successful in a given immune environment, and particularly for sustaining long-term infection. The narrow post-switch vulnerability window we describe here is not the only mechanism affecting VSG diversity, but it may prove to be an important contributing factor. Ultimately, the question of whether a more stringent immune evasion threshold would permit (or disallow) sustained infections cannot be answered with certainty.

A prior study by Dubois et al.[27] examined the phenomenon of VSG coat replacement from another angle, demonstrating that trypanosomes simultaneously expressing two VSGs elicit a weaker Ab response than trypanosomes expressing only one VSG. This result informs the effect of VSG coat replacement on the development of new Abs targeting subsequent waves of parasitemia, but does not address our focus, the effect of the previously stimulated Ab response on the switched (and partially coat-replaced) trypanosomes themselves. We did not examine the Ab response elicited by our DE clones in detail, as it was beyond the scope of our study. However, we do not see any conflict between the results of the two studies and suggest that they may be viewed as complimentary analyses.

The reasoning behind our focus on the role of IgM, as opposed to other components of anti-trypanosomal immunity, warrants some additional explanation. While innate immune factors affect host susceptibility to *T. brucei* and can exert control on the progression of infection, they do so only in a non-VSG-specific manner[28–30]. Innate factors should therefore have a similar effect on all clones in our immunization-challenge experiments, and should not significantly contribute to the clone-dependent immune evasion patterns observed. Abs are the key components in interactions involving antigenic variation, as they are the only immune factors that recognize specific VSGs and mediate clearance of peaks of parasitemia[29]. Of all the immunoglobulin (Ig) classes, multiple studies have provided direct evidence of a dominant role of IgM in trypanosome clearance[6, 19]. Furthermore, the rough 5–8-day periodicity of parasitemia waves[3, 4, 7] aligns with the expected time frame of successive IgM responses. Still, it should be noted that the importance of IgM in trypanosome clearance has been the subject of some controversy. Experimental infections in IgM-deficient (but IgD-expressing, and thus also termed "IgD-only") mice[31] have shown mild phenotypes in parasitemia progression as compared to infections in wild-type mice, prompting the conclusion that IgG may be more responsible for parasite suppression[29, 32]. However, the IgM-deficient mice generated compensatory IgD Abs (normally absent in wild-type mice) with the same dynamics as a typical IgM response. Thus, an alternative explanation for these results is that trypanosome clearance is mediated by the earliest arising Ab response, and that IgD is largely capable of performing this function in place of IgM. For all of these reasons, we maintain that IgM is the Ig class most relevant to cells undergoing VSG coat replacement.

Our mathematical model of the IgM–VSG coat interface (Fig. 4b, c) provides a framework for evaluating the impact of VSG density on IgM–VSG binding based on the two-dimensional scale of these interactions, but we must additionally consider the dynamic nature of the system. The VSG coat is highly fluid, and each monovalent IgM–VSG interaction has some discrete duration determined by the IgM's binding kinetics. Thus, the ability of a single IgM to make contacts with multiple cognate VSGs is actually based on the likelihood of a second cognate VSG coming into range of a given IgM during an initial monovalent interaction. This likelihood is highly influenced, but not fully determined, by average VSG spacing. Further, hydrodynamic flow generated by beating of the trypanosomal flagellum has been demonstrated to push surface VSG–Ig complexes toward the posterior end of the

cell[33], which could produce a gradient of increasing VSG density on the trypanosome surface in vivo. This process may thereby increase the likelihood of polyvalent IgM–VSG interactions, and consequently decrease the expected threshold amount of cognate VSG necessary for IgM binding to a lower level than our model predicts. Conversely, the shear forces that hydrodynamic flow applies to IgM–VSG interactions could decrease the duration of each monovalent interaction, producing the opposite effect on our expected threshold amount. Overall, in cases where trypanosomes have ≥7.6% initial VSG remaining, a single IgM is likely to encounter multiple initial VSGs regardless of whether the VSG coat is in motion. For lower VSG percentages, the expected effect is more uncertain, though our empirical evidence suggests that our DE-1.3 clone is indeed below the IgM binding threshold (Fig. 4a). A deeper mechanistic evaluation would require performing a similar analysis of IgM binding to trypanosomes in whole blood under flow conditions, but the technical capability to perform such experiments does not exist, at present.

Finally, in addition to addressing a gap in our understanding of *T. brucei* pathogenesis, the work presented here may hold broader relevance to the basic biology of IgM. The fact that IgM binding affinity is dependent on antigen density has been known for many years[21], but the study presented here is the first, to our knowledge, in which antigen has been titrated on the surface of a pathogen, and a relatively sharp, density-dependent cutoff of IgM functionality has been established in vivo. Both induced IgM responses and natural IgMs (non-mutated IgMs constitutively produced in the absence of immune stimulation) form a crucial first line of defense against many other pathogens[34, 35] as well as pre-cancerous cells[36, 37]. Further antigen density analyses may reveal other clinically relevant IgM binding thresholds, or assist in determining which specific antigens are conducive to IgM targeting in cases where these are unknown.

## Methods

**Trypanosome cell lines and mouse strains**. All trypanosome cell lines used in this study were bloodstream-form trypanosomes ultimately derived from the *Lister427* "single marker" (SM) cell line[38]. Trypanosomes were cultivated in HMI-9 medium with 10% fetal bovine serum (Sigma F4135) at 37 °C and 5% $CO_2$. "WT-VSG2" trypanosomes are unaltered SM cells. Derivation of the Δ70 clone has been described previously[13]. The WT-VSG3 clone was isolated from a group of switched cells from the 70.II line published by Hovel-Miner et al.[13]. In this clone, *VSG3* is the active *VSG* gene expressed from Bloodstream Expression Site 1 (BES1). The Δ70 and WT-VSG3 clones were provided by Dr. Galadriel Hovel-Miner (current address: George Washington University). VSG2-expressing trypanosomes lacking GPI-PLC (*GPI-PLC* $^{-/-}$)[39] were provided by Dr. George Cross (Rockefeller University). All mice were female, wild-type C57BL/6J, aged 6–9 weeks at experiment start (Jackson Laboratory). All animal experiments were approved by Rockefeller University's institutional animal care and use committee under protocol #16894.

**Engineering sortaggable VSG2 (VSG2$^{STa}$) and VSG2$^{STa}$ knock-in clones**. The amino-acid sequence GGGENLYFQGGGGGGG was cloned into VSG2 (GenBank X56762) replacing residues E29-G31[40]. This segment immediately follows two Ala residues that form the N terminus of the mature VSG2 protein after the preceding leader peptide sequence is removed[41] (thus the mature VSG2$^{STa}$ protein begins AAGGGENLYFQGGGGGGGFKQAFWQPLCQVS-, insertion underlined). The full plasmid construct for VSG2$^{STa}$ knock-in (pSY37F1D-CTR-BSD), derived from pSY37[42] (a gift of Dr. Hee-sook Kim, Rockefeller University) contains, in order: 1156 bp of the "cotransposed region" (CTR) preceding the VSG2 open reading frame in BES1 of the *Lister427* genome, immediately followed by the previously described *VSG2$^{STa}$* gene, then a Blasticidin resistance gene (BSD), and finally a 200 bp telomere seed region. A BglII restriction site was used to linearize the plasmid, leaving the CTR and telomere seed (the sources of homologous recombination flanking VSG2 within the target genome) on opposing ends of the resulting DNA fragment. This fragment was then used for transfection of WT-VSG2 and Δ70 cells via the Amaxa nucleofector protocol as previously described[43], resulting in clones KI-VSG2$^{STa}$ and Δ70$^{STa}$, respectively. This transfection replaces the previously active *VSG2* gene, such that KI-VSG2$^{STa}$ and Δ70$^{STa}$ express only VSG2$^{STa}$.

**Sortagging reaction**. Sortase A expression and purification is described in Supplementary Methods. 5-FAM (5-carboxyfluorescein) or TAMRA-peptide conjugates ([5-FAM]/[TAMRA]-GGGSLPSTGG, sortase recognition motif

underlined) were synthesized by Elim Biopharmaceuticals. Sortagging solutions containing 100 μM purified Sortase A and 300–600 μM fluorophore-peptide [Concentration varied between experiments of different types, but was consistent for replicates of the same experiment] in HMI-9 media were incubated on ice for 30–60 min. Trypanosomes expressing VSG2$^{STa}$ were pelleted, resuspended in the sortagging solution, and incubated for 1 h at 4°C on an inversion rotator. Cells were then pelleted, washed once with HMI-9 media, and pelleted again before final resuspension in HMI-9. These cells were then analyzed immediately or returned to normal culture conditions. For Fig. 1b, Sortagged KI-VSG2$^{STa}$ cells were cultured for 12 h prior to imaging. Imaging was performed using an ImageStream-X flow cytometer (Amnis). For Fig. 1c, sortagged KI-VSG2$^{STa}$ cells were isolated from culture at indicated time points and analyzed via BD-FACSCalibur flow cytometer.

**Switch induction for Δ70 and Δ70$^{STa}$ cell lines**. Δ70 or Δ70$^{STa}$ trypanosomes were diluted to $10^5$ cells/ml in culture flasks containing HMI-9 with 1 μg/ml Doxycycline (Dox), then incubated in normal culture conditions for 6 h. At 6 h, cells were pelleted, washed with Dox-free HMI-9, then resuspended in Dox-free HMI-9 and incubated again in normal culture conditions. Non-induced cells were treated identically, except without Dox in the initial 6 h incubation. For sortagging/switching experiments (Fig. 1d, e), Δ70$^{STa}$ cells were sortagged as described above (using 600 μM 5-FAM-peptide) immediately prior to switch induction.

**Monoclonal anti-VSG antibodies**. The monoclonal anti-VSG antibodies used in this study were produced by the Antibody and Bioresource Core Facility of the Memorial Sloan Kettering Cancer Center. The anti-VSG2 antibody is clone VSG221-31G9, while the anti-VSG3 antibody is clone VSG224-11D6 (both publicly available http://macfwebext.mskcc.org/EstablishedMAb/SearchEstablishedMAbs.aspx). Only these two clones were used, but experiments were performed using individually prepared antibody-fluorophore conjugates (the fluorophore conjugates used in each experiment are indicated in Methods and figure legends). These antibody-fluorophore conjugates were generated using purified antibody and Invitrogen protein labeling kits, following manufacturer instructions (Alexa647 A20173; Alexa488 A10235; Pacific Blue P30012; FITC F10240).

**Flow cytometry sample preparation**. All of the following samples, solutions, and incubation steps were at 4 °C or on ice to prohibit antibody internalization during sample preparation. Samples of $2 \times 10^6$ trypanosomes were isolated from induced or non-induced cultures, pelleted, and resuspended in HMI-9 containing monoclonal antibodies recognizing VSG2 (the pre-switch VSG, antibody was anti-VSG2-Alexa647 [1:2000 dilution]) and VSG3 (one of many post-switch VSGs, antibodies were anti-VSG3-Alexa488 [1:400 dilution] or anti-VSG3-Pacific Blue [1:100 dilution]). Samples were incubated on an inversion rotator for 10 min, then pelleted and washed once with HMI-9. Samples were then pelleted again, resuspended in HMI-9 containing 500 ng/ml Propidium Iodide (BD 556463), and analyzed using a BD-FACSCalibur (Fig. 1a; Supplementary Fig. 1a) or BD-LSRII (Fig. 1d, e; Supplementary Fig. 1c) flow cytometer. Single-stain samples for flow cytometry compensation were prepared identically using the same antibody and Propidium Iodide concentrations or sortagging procedures.

**Conversion of MFI values to MESF values**. Quantum FITC and Alexa647 MESF bead kits (Bangs Laboratories cat. 555 and 647) were used to convert sortag (FITC) or anti-VSG2-antibody stain (Alexa647 conjugate,1:2000 dilution) raw mean fluorescence intensity (MFI) values to absolute Molecules of Equivalent Soluble Fluorochrome (MESF) units[44] per manufacturer instructions. For each fluorophore, a set of Quantum beads were run on the flow cytometer immediately prior to trypanosome sample acquisition, using identical settings. These bead sets consisted of five or six samples of beads, each of which were conjugated to a known amount of fluorophore. Bead MFI values were then inputted into Bangs Laboratories software provided with the kits, which generated a calibration curve for converting MFI of other samples to MESF units. MFI values from trypanosome samples were then inputted into this same software to obtain absolute MESF values.

**VSG $t_{E1/2}$ calculations**. Flow cytometry data were analyzed using FlowJo software. Trypanosome populations were gated via forward and side scatter, then dead cells were excluded by gating cells negative for Propidium Iodide staining. $t_{E1/2}$ values were calculated from data from the 12 h post-induction time point onward, because the presence of cells expressing switched VSGs (see VSG3-positive cells in Fig. 1a) indicated that the genetic VSG switch had occurred by this point, and thus VSG coat replacement was actively taking place. "Switcher" populations were gated from "non-switcher" populations by anti-VSG2 antibody staining (Alexa647 conjugate,1:2000 dilution; see Supplementary Fig. 1c) for 24–48 h time points. For the 12 h time point, the switcher and non-switcher populations were not gated separately, because at this early stage the switchers had not lost enough VSG221 to be identified as distinct from the non-switchers. MFI values from switcher and non-switcher populations were converted into MESF units as described in the previous section, and $t_{E1/2}$ values were calculated via non-linear regression analysis using Prism software.

**CFSE cell proliferation analysis**. Immediately following sortagging reaction (described above), trypanosomes were pelleted and resuspended in CFSE buffer (PBS, 0.1% BSA, 154 mM Ggucose), then added to solutions containing CFSE buffer with 800 nM (final concentration) CFSE (Invitrogen C34554). Samples were incubated for 15 min at 37°C, then the reaction was quenched by adding the entire volume to 5× volume of HMI-9 and incubating for 5 min at room temperature. Cells were pelleted and washed once with HMI-9, then induced to switch as described above or returned to normal culture conditions. Sample collection and preparation, and flow cytometry gating were performed as described above.

**Generation of DE clones**. Simultaneous expression of two *VSGs* from the active Bloodstream Expression Site (BES) has been described previously[15]. For our constructs, a ~1 kb stretch of unique intergenic region (UIR) from *Lister427* BES1 was amplified from WT-VSG2 genomic DNA (gDNA) using primers JPC1_UniqueFor and JPC2_UniqueRev, while a pUC19 backbone was amplified using primers JPC3_pUC19For and JPC4_pUC19Rev (see primer sequences in Supplementary Methods). These PCR fragments were digested with XhoI and BglII and ligated to create plasmid pJP01. The UIR contains a BsaAI blunt end restriction site near the center of the region that was used for subsequent cloning steps. pJP01 also contains NotI restriction sites flanking the UIR, which allow release of this fragment.

For each DE clone construct, the *VSG2* gene, its flanking UTRs, and a Blasticidin (BSD) resistance cassette were cloned into the pJP01 vector (cut at the BsaAI site in the center of the UIR) by Gibson assembly (cloning kit: NEB E5510S) using PCR fragments with 30 bp of overlapping homology at the junctions. The resulting plasmids were digested with NotI and transfected into parental clone WT-VSG3 via the Amaxa nucleofector protocol as previously described[43]. Cloning details for each specific clone and primer sequences can be found in Supplementary Information.

**Analysis of VSG2 surface expression levels of DE clones**. VSG2 and VSG3 surface expression were measured by quantitative flow cytometry analysis. Flow cytometry samples were prepared as described above, with the following exceptions: Samples for Supplementary Fig. 2b, c were prepared using single anti-VSG3 antibody solutions diluted as indicated. For Fig. 2b and Supplementary Fig. 2a, samples contained $3 \times 10^4$ cells of the indicated clone, pelleted along with $10^6$ WT-VSG3 cells. This permitted use of a saturating anti-VSG2-Alexa647 antibody concentration (1:100 dilution, see Supplementary Fig. 2a), because fewer VSG2-expressing cells were present to draw antibody out of solution. The additional WT-VSG3 cells allowed effective pelleting of such small samples of the VSG2-expressing clones. Saturating levels of anti-VSG2-Alexa647 antibody were necessary to ensure that limiting antibody concentration did not skew relative surface VSG2 expression measurements.

For Fig. 2b, Conversion to MESF units was performed as described above to convert raw fluorescence intensity data to absolute values. Adjusted DE clone and WT-VSG2 MESF values were calculated by subtracting background MESF (MESF value of WT-VSG3 cells) from the raw MESF values of these clones, and VSG2 percentages were then calculated by dividing the adjusted DE clone MESF values by the adjusted WT-VSG2 MESF value. MESF calculation was not performed for Fig. 2c, because relative staining levels were sufficient to assess VSG2 expression stability over time. Forward and side scatter, and dead cell exclusion gating were identical to that described above. All samples for Fig. 2 and Supplementary Fig. 2 were analyzed using a BD-LSRII flow cytometer.

**Immunization of mice with UV-irradiated *GPI-PLC$^{-/-}$* trypanosomes**. VSG2-expressing *GPI-PLC$^{-/-}$* trypanosomes were pelleted from culture, washed with irradiation buffer (PBS, 154 mM glucose), and resuspended in irradiation buffer to a density of $10^7$ cells/ml. A volume of 1 ml of this resuspension was aliquoted into each well of a six-well tissue culture plate (Falcon 353046). Plates were UV-irradiated for 10 min in 1 min intervals using a FB-UVXL-1000 UV crosslinker (Fisher Scientific). Plates were swirled between 1 min intervals to ensure equal irradiation of trypanosomes. Irradiated cells were pelleted and resuspended in HMI-9 at a concentration of $15 \times 10^6$ cells/ml. A volume of 100 µl of this solution ($3 \times 10^6$ trypanosomes) were injected i.p. into mice. This immunization procedure was performed twice on each mouse (day 0 and day 3). For live trypanosome comparison immunizations (Supplementary Fig. 3a), mice were infected on day 0 with 100 or 1000 live WT-VSG2 trypanosomes via i.p. injection. Infections were cleared with 250 ng berenil/mouse injected i.p. on the day trypanosomes were identified by blood smear (day 4 for 1000 injected, day 5 for 100 injected). Berenil treatment was repeated 24 h after initial treatment. Serum samples for ELISA analysis were obtained via submandibular bleed, and serum was separated from whole blood using Microtainer serum collection tubes (BD 365967).

**Immunization-challenge experiments**. Mice were immunized with UV-irradiated GPI-PLC$^{-/-}$ trypanosomes, and serum samples were obtained for ELISA analysis (from immunized and non-immunized control mice) on day 8 post immunization as described above. Later that same day, mouse groups were challenged via i.p. injection of 100 live WT-VSG2, WT-VSG3, DE-11, DE-7.6, or DE-1.3 trypanosomes in HMI-9. Mice were monitored for survival twice daily on days 4–16 post challenge, then once daily until day 23. Mice were also monitored for trypanosome infection by blood smear (blood from mouse tail clippings was examined by eye

under a microscope for the presence of trypanosomes), and permanently scored as "confirmed infected" once trypanosomes were identified, regardless of whether parasitemia subsequently fell to undetectable levels. Blood smear monitoring occurred daily from days 4–8 post challenge (i.e., through the first peak of infection) for all mice. After day 8, all mice that had not yet exhibited parasitemia were monitored daily until the mouse died or the experiment ended. At all times, if blood smears indicated a quantifiable level of parasitemia, parasitemia was measured using a hemocytometer. One mouse in the non-immunized DE-11 group was excluded from analysis because ELISA results showed high anti-VSG2 IgM titers despite lack of immunization, and one mouse in the immunized DE-7.6 group was excluded because it died during the course of the experiment from a cause unrelated to trypanosomiasis. We did not use a statistical method to estimate sample size, as we did not have a pre-specified effect size. We initially estimated that five mice/group would be enough to identify a result and repeated that number in subsequent experiments to ensure reproducibility. We assumed mice delivered from Jackson laboratory were sufficiently randomized upon arrival and did not further randomize the mice into experimental groups. Investigators were not blinded to experimental groups, but metrics used for these experiments (assessment of survival, identification of parasites, parasitemia counts) were not subjective.

**ELISA assays**. ELISA assays were performed using established methods. ELISA plates (Corning 9018) were coated with 2.3 µg/ml purified VSG2 in borate buffer (100 mM boric acid, 25 mM sodium borate, 75 mM sodium chloride), then blocked with blocking buffer (1% BSA in borate buffer). Primary antibody solutions contained 1:30 (Fig. 3a) or 1:100 (Supplementary Fig. 3a) dilutions of mouse antisera in blocking buffer. Secondary antibody solutions contained 1:1000 dilutions of Goat Anti-Mouse IgM-AP (SouthernBiotech 1021-04) or Goat Anti-Mouse IgG1-AP (SouthernBiotech 1070-04). Washes were performed using PBS with 0.05% Tween 20. Wells were developed in developing buffer (0.1 M glycine, 1 mM zinc chloride, 1 mM magnesium chloride) with 1 mg/ml PNPP substrate (ThermoFisher 34047) and OD 405 nm was read using a Spectramax M3 plate reader (Molecular Devices).

**Elicitation of VSG2-specific polyclonal IgM antisera**. Two mice were injected i.p. with $1–2 \times 10^4$ WT-VSG2 trypanosomes in HMI-9. After 4 days, infections were cleared with 250 ng berenil/mouse injected i.p., and berenil treatment was repeated after 24 h. On day 8 post injection, blood was collected via cardiac puncture, and serum was separated from whole blood using Microtainer serum collection tubes (BD 365967).

**IgM binding assay**. Flow cytometry samples were prepared as described above (Flow cytometry-based VSG coat replacement analyses), except antibody binding was done in two steps: (1) VSG2-specific polyclonal IgM antisera, diluted 1:100 in mouse serum (Mouse Reference Serum, Bethyl Laboratories, RS10-101). (2) 1:500 dilution of Goat Anti-Mouse IgM-FITC (SouthernBiotech 1021-02) in HMI-9. Samples were incubated for 10 min in each antibody solution and washed once with HMI-9 after each binding step. Control samples labeled with monoclonal anti-VSG2-Alexa647 were prepared using a single antibody incubation step. Forward and side scatter, and dead cell exclusion gating were identical to that described above.

**Development of mathematical IgM–VSG coat binding interface model**. To approximate the binding "footprint" of a pentameric IgM molecule on the VSG coat, we first modeled the VSG coat as a regularly spaced hexagonal grid, as has been shown in previous studies[24]. A regular hexagon of height $h$ has six sides of length equal to $(h/3^{1/2})$. A hexagon with a height of 40 nm (the upper estimate of the diameter of an IgM molecule) thus has sides of length ~23.1 nm. If such a hexagon has five VSG homodimers on each side, they will be equally spaced at ~5.8 nm (23.1 nm/4 inter-homodimer spaces), a value extremely close to previously estimated homodimer spacing (5.7 nm spacing with a random displacement up to 0.5 nm)[24]. Filling this hexagon with additional VSG homodimers at the same spacing, we estimated that 61 VSG homodimers (or 122 VSG monomers) would be within reach of a single IgM with a 40 nm diameter (Fig. 4b, c). The same calculations were performed for IgM with a 30 nm diameter (Supplementary Fig. 4): 30 nm/3$^{1/2}$ gives a side length of ~17.3 nm. Assuming four VSG homodimers on each side, they will again be equally spaced at ~5.8 nm (17.3 nm/3 inter-homodimer spaces). Filling this hexagon results in 37 VSG homodimers (or 74 VSG monomers) within reach of a single IgM with a 30 nm diameter.

**Data availability**. All data (and reagents) that support the findings of this study are freely available from the corresponding author upon request.

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

## Acknowledgements

We thank George A.M. Cross for his invaluable advice and support, Galadriel Hovel-Miner and Heesook Kim for allowing us to use reagents prior to publication, and Svetlana Mazel, along with the rest of the Rockefeller University Flow Cytometry Resource Center staff, for advice and support. This work has been supported by Rockefeller University Graduate School funds to J.P. and by NIH/NIAID (grant #AI085973) to F.N.P.

## Author contributions

All experiments presented were designed by J.P. and F.N.P., and performed by J.P. (Figs. 1, 2 and 4; Supplementary Figs. 1, 2 and 4), and by J.P. together with S.C. (Fig. 3; Supplementary Fig. 3). The manuscript was written by J.P. and F.N.P.

## Additional information

**Competing interests:** The authors declare no competing financial interests.

