## [Peer Review File · Nature Communications]

Reviewers' Comments:

Reviewer #1:

Remarks to the Author:

The manuscript by Papavasiliou and colleagues reports the unexpected finding that, during an antigenic switch, trypanosomes are vulnerable for a much short period of time than previously anticipated. This is a very important finding now that several groups in the field are beginning to study antigenic variation using animal models. Although many studies have been published characterizing the immune response against *Trypanosoma brucei*, very little remains known about the interactions at the actual interface pathogen-antibody. This is an important concept that interests not only Trypanosome community, but also immunologists.

To tackle this difficult problem, the authors developed novel, clever and quantitative experimental approaches, including a mathematical model at the end.

The work described in this manuscript is highly original and convincing. There are a number of small points that could benefit from attention.

FIGURE 1

Although Figure S1 suggests that there are no major differences in replication between switchers and non-switchers, could the authors show/comment on whether there are any differences in the growth rate of cells with no tag in N-terminus, with the Tag and Tag+Sortase-Fluorophore?

FIGURE 2

- It is not entirely clear how the authors converted raw fluorescence intensity into absolute units.
- Did the authors check if the VSG staining was made in saturation conditions (i.e. that the antibody is in excess)? If not, it could mean that the 100% in WT-221 was probably underestimated and, consequently, the expression of VSG221 in Double Expressors was overestimated. Could the authors show, in supplementary material, that with 5-fold, 50-fold or 500-fold more VSG221 antibody, for instance, they would obtain the same relative expression numbers as indicated in panels B and C?
- Did the authors check the impact of VSG221 expression on VSG3 levels? It seems there are contradicting data in the field suggesting that the "original" VSG is either reduced or remains unchanged.
- Is the key of panel C correct? Should it be DE-1.3 and DE-11 (instead of DE-1.4 and DE-10), like in panel B?

FIGURE 3

- To show the importance of IgM versus IgG, could the authors consider an experiment similar to Figure 3, but in which an extra group is immunized mice with VSG221 purified protein? Shouldn't this trigger an IgG response? Based on the authors' model, when these mice are challenged, the mixed cover parasites should be susceptible to elimination for a longer period of time than in mice immunized with irradiated parasites.

SUPPLEMENTARY TABLE 1

Could the authors explain the equations in supplementary table 1 to a "biologist" reader?

FIGURE 4

Could the authors describe in more detail how the model was done?

DISCUSSION

The authors cite and discuss that Dubois et al. have a complementary report to theirs. In my opinion, the work from Pinger et al. is, biologically more relevant because a mammalian host is normally infected by parasites that have a single VSG cover and not by parasites that have a

mixed coat. As far as we know, the immune system does not have to simultaneously generate an immune response to two "new" VSGs at the surface of the same cell (as addressed in Dubois et al). Rather, the host generates a second immune response AFTER having established a first one earlier. The authors should, therefore, emphasize that their work is not simply "a complementary analyses", but it is likely more relevant.

STATISTICS

It is not entirely clear from the figure legends how many times the experiments were performed, the number of independent clones, etc. I trust Nature now has personnel that will thoroughly scrutinize these details.

Reviewer #2:

Remarks to the Author:

The cell surface of African trypanosomes is covered by a VSG coat that is the modulator of antigenic variation. An old question is what happens to the old VSG when the trypanosomes turns off the old VSG gene and starts transcribing a new VSG gene. There are $\sim 5 \times 10^6$ VSG molecules on the cell surface, representing $\sim 10\%$ of total cell protein. There is a discrete time needed to replace this old VSG but it is unknown whether it occurs by simple growth and dilution or whether the old VSG is removed more rapidly by an unknown process.

The manuscript describes three sets of experiments.

In the first (Figure 1) flow cytometry and specific tagging of one VSG are used to show that the old VSG is lost by growth and dilution. These data are convincing as they stand. It may be possible to improve the results by including the growth curves for the cell lines. For example in figure 1c there could be a plot of cumulative cell number x mean fluorescence against time would be more informative?

In the second (Figures 2 and 3), transgenic cell lines expressing two VSGs at a range of ratios were used to infect mice that had or had not been pre-immunised to the minor VSG. It is known that pre-immunisation with irradiated trypanosomes triggers an IgM response. The data indicated that trypanosomes expressing $> 7.6\%$ of a minor VSG were cleared but those expressing 1.3% were not. This is a very interesting observation.

In the third set of experiments (figure 4), binding of IgM to transgenic cell lines expressing two VSGs at a range of ratios was investigated. There was almost undetectable binding to trypanosomes expressing 1.3% minor VSG. Again, these observations are very interesting. The model/discussion of why IgM does not bind to cells expressing the minor VSG at 1.3% misses out a couple of important. First the VSG is very mobile in the plane of the membrane and the degree of avidity will not be determined by mean VSG spacing but by a combination of monovalent off time (rather than affinity); the time taken for a second VSG to come within range for a second interaction and the effect of hydrodynamic flow on orientation forced upon the VSG IgM and the movement of the complex towards the posterior of the cell. Bearing in mind the latter it may be informative to do the IgM binding experiments in serum not HMI-9 medium.

Overall, this is elegant work and the data largely justify the conclusions.

Very Minor points

There are 5×10^6 VSGs on the cell surface, each dimer is one VSG molecule.

The inducible VSG switching line is a transgenic line not a mutant

The low switch rates are in lab adapted cell lines not wild ones

Point by Point Response to Reviewers' comments:

Reviewer #1 (Remarks to the Author):

The manuscript by Papavasiliou and colleagues reports the unexpected finding that, during an antigenic switch, trypanosomes are vulnerable for a much shorter period of time than previously anticipated. This is a very important finding now that several groups in the field are beginning to study antigenic variation using animal models. Although many studies have been published characterizing the immune response against *Trypanosoma brucei*, very little remains known about the interactions at the actual interface pathogen-antibody. This is an important concept that interests not only the Trypanosome community, but also immunologists.

To tackle this difficult problem, the authors developed novel, clever and quantitative experimental approaches, including a mathematical model at the end.

The work described in this manuscript is highly original and convincing. There are a number of small points that could benefit from attention.

We thank the reviewer for their kind comments.

FIGURE 1

Although Figure S1 suggests that there are no major differences in replication between switchers and non-switchers, could the authors show/comment on whether there are any differences in the growth rate of cells with no tag in N-terminus, with the Tag and Tag+Sortase-Fluorophore?

We have included this information in the manuscript text (p.4, paragraph 1), and to assess this to completion we have performed a straightforward CFSE dilution experiment (with or without the sortag). This is now provided as a new panel in Supplementary figure 1 (Fig. S1c).

FIGURE 2

- It is not entirely clear how the authors converted raw fluorescence intensity into absolute units.

This is now explained in detail in the text (see legend to Figure 1, and methods).

- Did the authors check if the VSG staining was made in saturation conditions (i.e. that the antibody is in excess)? If not, it could mean that the 100% in WT-221 was probably underestimated and, consequently, the expression of VSG221 in Double Expressors was overestimated. Could the authors show, in supplementary material, that with 5-fold, 50-fold or 500-fold more VSG221 antibody, for instance, they would obtain the same relative expression numbers as indicated in panels B and C?

We were concerned about this issue as well, and had performed an anti-VSG2 antibody titration experiment as a preamble to selecting the concentration used for these measurements. This data is now included as a panel in a new supplementary figure (Figure S2).

- Did the authors check the impact of VSG221 expression on VSG3 levels? It seems there are contradicting data in the field suggesting that the "original" VSG is either reduced or remains unchanged. We have analyzed the levels of VSG3 expressed on the surface of WT-VSG3 cells and each of the DE (VSG2 co-expressing) clones via anti-VSG3 antibody staining, and found that they each express approximately equal VSG3 amounts. Thus, the additional expression of VSG2 does not appear to reduce the amount of VSG3 present at the cell surface. These data are now included and discussed in Supplementary figure 2.

- Is the key of panel C correct? Should it be DE-1.3 and DE-11 (instead of DE-1.4 and DE-10), like in panel B?

We thank the reviewer for catching this, which has now been corrected!

FIGURE 3

- To show the importance of IgM versus IgG, could the authors consider an experiment similar to Figure 3, but in which an extra group is immunized mice with VSG221 purified protein? Shouldn't this trigger an IgG response? Based on the authors' model, when these mice are challenged, the mixed cover parasites should be susceptible to elimination for a longer period of time than in mice immunized with irradiated parasites.

We have considered this experiment carefully, and believe that it is unlikely to provide unequivocally interpretable data for the following reasons:

1) To exclusively assess the ability of IgG antibodies to clear trypanosomes with dilute surface VSG2, the experiment would ideally be performed on IgG-only mice, however those do not exist. The less-optimal version is the one the reviewer suggested, namely, immunization with VSG protein, followed by boosts with additional protein, until we generate a response that is dominated by IgG. We have attempted this immunization and found that generation of a strong IgG response requires a minimum of ~3-4 weeks, and at that point mice still retain high IgM titers in addition to IgG. We have not been able to extinguish the IgM response

2) An additional and major confounding factor introduced by this approach is that IgM antibodies are also capable of undergoing affinity maturation over time. Thus, any possible trypanosome clearance at that stage could be a result of the presence of IgG antibodies, or of the increased clearing capabilities of affinity-matured IgM. We believe that this issue also skews the experiment further from physiological relevance, since during trypanosome infections peaks of parasitemia are cleared very quickly (within ~7-8 days) by early antibody responses, which have undergone little, or no affinity maturation.

3) We have also considered other possible experiments, such as infusion with anti-VSG IgM or anti-VSG IgG prior to or concurrent with infection with the cognate trypanosome. However, these seem increasingly artificial and also produce a number of experimental design questions without straightforward answers (e.g. at what time point(s) following immunization would the IgG be collected to provide appropriate physiological relevance?).

Ultimately, while we agree that the protective contribution of anti-VSG IgG warrants future research (in relation to our model or other aspects of anti-trypanosomal immunity, such as VSG-specific memory responses) we hope the reviewer may be satisfied with this explanation as to why we did not pursue this experimental route within this current body of work. .

SUPPLEMENTARY TABLE 1

Could the authors explain the equations in supplementary table 1 to a "biologist" reader?

We now detail these in the legend to supplementary table 1.

FIGURE 4

Could the authors describe in more detail how the model was done?

We now describe this in detail in the methods section (p. 25 of the manuscript)

DISCUSSION

The authors cite and discuss that Dubois et al. have a complementary report to theirs. In my opinion, the work from Pinger et al. is, biologically more relevant because a mammalian host is normally infected by parasites that have a single VSG cover and not by parasites that have a mixed coat. As far as we know, the immune system does not have to simultaneously generate an immune response to two "new" VSGs at the surface of the same cell (as addressed in Dubois et al). Rather, the host generates a second immune response AFTER having established a first one earlier. The authors should, therefore, emphasize that their work is not simply "a complementary analyses", but it is likely more relevant.

While we are in agreement with the reviewer, we would hesitate to make such a strong statement. We prefer that the readers make their own conclusions (as the reviewer has already done).

STATISTICS

It is not entirely clear from the figure legends how many times the experiments were performed, the number of independent clones, etc. I trust Nature now has personnel that will thoroughly scrutinize these

details.

We now provide these data both to the journal "checklist" and also explicitly in the figure legends and other appropriate spots within the paper.

Reviewer #2 (Remarks to the Author):

The cell surface of African trypanosomes is covered by a VSG coat that is the modulator of antigenic variation. An old question is what happens to the old VSG when the trypanosomes turns off the old VSG gene and starts transcribing a new VSG gene. There are $\sim 5 \times 10^6$ VSG molecules on the cell surface, representing $\sim 10\%$ of total cell protein. There is a discrete time needed to replace this old VSG but it is unknown whether it occurs by simple growth and dilution or whether the old VSG is removed more rapidly by an unknown process.

The manuscript describes three sets of experiments.

In the first (Figure 1) flow cytometry and specific tagging of one VSG are used to show that the old VSG is lost by growth and dilution. These data are convincing as they stand. It may be possible to improve the results by including the growth curves for the cell lines. For example in figure 1c there could be a plot of cumulative cell number x mean fluorescence against time would be more informative?

Figure 1 now includes these data as requested by the reviewer.

In the second (Figures 2 and 3), transgenic cell lines expressing two VSGs at a range of ratios were used to infect mice that had or had not been pre-immunised to the minor VSG. It is known that pre-immunisation with irradiated trypanosomes triggers an IgM response. The data indicated that trypanosomes expressing $> 7.6\%$ of a minor VSG were cleared but those expressing 1.3% were not. This is a very interesting observation.

In the third set of experiments (figure 4), binding of IgM to transgenic cell lines expressing two VSGs at a range of ratios was investigated. There was almost undetectable binding to trypanosomes expressing 1.3% minor VSG. Again, these observations are very interesting.

The model/discussion of why IgM does not bind to cells expressing the minor VSG at 1.3% misses out a couple of important factors. First the VSG is very mobile in the plane of the membrane and the degree of avidity will not be determined by mean VSG spacing but by a combination of monovalent off time (rather than affinity); the time taken for a second VSG to come within range for a second interaction and the effect of hydrodynamic flow on orientation forced upon the VSG-IgM and the movement of the complex towards the posterior of the cell.

We thank the reviewer for these comments. While we had thought about these exact issues, we did not detail them in the text, but have done so now in the revised Discussion.

Bearing in mind the latter it may be informative to do the IgM binding experiments in serum not HMI-9 medium.

We have performed the experiment as suggested. We report the result in figure 4a, replacing the original panel (which looked nearly identical). The methods have been fixed to reflect the change as well. Additionally, we now discuss the possibility that binding could be affected by the presence of not only serum, but also of whole blood and hydrodynamic flow, in the revised Discussion.

Overall, this is elegant work and the data largely justify the conclusions.

We thank the reviewer for their comments and overall input.

Very Minor points

There are 5×10^6 VSGs on the cell surface, each dimer is one VSG molecule.
The inducible VSG switching line is a transgenic line not a mutant
The low switch rates are in lab adapted cell lines not wild ones

These issues have now been fixed, excepting the point on VSGs existing as dimers. We are hesitant to fully apply the reviewer's suggestion here, as we have additional structural data (detailed in a separate manuscript) indicating that some VSGs may not exist in dimeric form.

Reviewers' Comments:

Reviewer #1:

Remarks to the Author:

The authors have amply addressed all my questions and comments. Specifically, I agree with the arguments relative to the importance of IgM and IgG (Figure 3).

Reviewer #2:

Remarks to the Author:

The authors have addressed the points I raised and I would recommend publication. The work represents a major advance in our understanding of how trypanosomes evade the adaptive immune response.